# Same Content, Different Representations: A Controlled Study for Table QA

**Yue Zhang**[1]* **Seiji Maekawa**[2] **Nikita Bhutani**[2]
[1]The University of Texas at Dallas   [2]Megagon Labs
yxz230014@utdallas.edu  {seiji,nikita}@megagon.ai

## Abstract

Table Question Answering (Table QA) in real-world settings must operate over both structured databases and semi-structured tables containing textual fields. However, existing benchmarks are tied to fixed data formats and have not systematically examined how representation itself affects model performance. We present the first controlled study that isolates the role of table representation by holding content constant while varying structure. Using a verbalization pipeline, we generate paired structured and semi-structured tables, enabling direct comparisons across modeling paradigms. To support detailed analysis, we introduce REPAIRTQA, a diagnostic benchmark with splits along table size, join requirements, query complexity, and schema quality. Our experiments reveal consistent trade-offs: SQL-based methods achieve high accuracy on structured inputs but degrade on semi-structured data, LLMs exhibit flexibility but reduced precision, and hybrid approaches strike a balance, particularly under noisy schemas. These effects intensify with larger tables and more complex queries. Ultimately, no single method excels across all conditions, and we highlight the central role of representation in shaping Table QA performance. Our findings provide actionable insights for model selection and design, paving the way for more robust hybrid approaches suited for diverse real-world data formats.

**Code** https://github.com/megagonlabs/RePairTQA

## 1 Introduction

Tables are a fundamental medium for storing and communicating information across domains such as finance (Chen et al., 2021), scientific communication (Ghosh et al., 2024), biomedical records (Ghosh et al., 2024), and the web (Chakrabarti et al., 2020). Unlocking the knowledge contained in these tables has motivated extensive research on Table Question Answering (Table QA), where models answer natural language queries grounded in tabular data (Pasupat & Liang, 2015; Iyyer et al., 2017). In practice, tables appear in both structured formats with rigid schemas and executable SQL queries (Zhong et al., 2017; Li et al., 2023; Wu et al., 2025a), and semi-structured formats where columns are irregular and cells contain free text (Chen et al., 2021; 2020; Singh et al., 2025). Structured tables enable precision and symbolic reasoning, while semi-structured tables offer robustness to noise and incomplete metadata. Since both formats coexist in real-world settings, understanding their impact on Table QA methods is crucial.

Despite substantial progress, this key question remains unanswered: how do different modeling paradigms handle variation in table representation? Existing benchmarks focus on query domain or complexity (Li et al., 2023; Wu et al., 2025a; Zhu et al., 2025), but fix the table format itself. As a result, models are optimized for a single representation, leaving their robustness to representation shift unclear. Current approaches fall into three main families: NL2SQL methods (Dong & Lapata, 2016; Zhong et al., 2017; Liu et al., 2022), LLM-based methods (Zhang et al., 2025c;a), and hybrid methods (Zhang et al., 2024; Ye et al., 2023; Abhyankar et al., 2025; Khoja et al., 2025). NL2SQL methods scale and reason precisely but are brittle under noisy or irregular schemas. LLM-based

---

*Work done during internship at Megagon Labs.

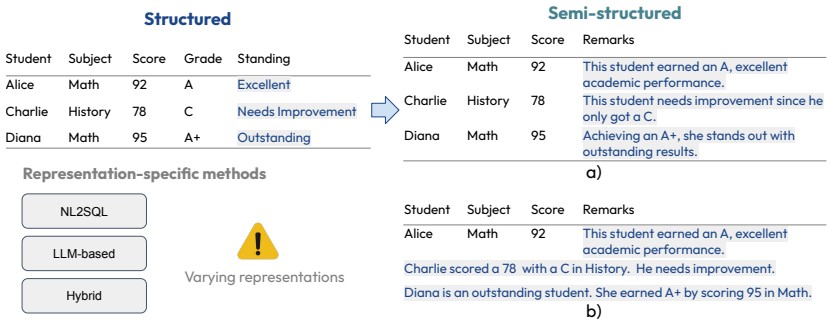

Figure 1: Structured vs. semi-structured formats of the same table pose challenges for Table QA methods that assume a fixed data format.

methods can handle semi-structured inputs containing free text but struggle on complex queries and long tables. Hybrid methods combine symbolic execution with neural reasoning to balance flexibility and precision. Without a controlled evaluation of representation effects, practitioners lack guidance on method selection, and researchers risk overfitting to narrow benchmark conditions.

To address this gap, we present the first controlled study of table representation in Table QA. Our framework generates paired structured and semi-structured tables using a verbalization pipeline, holding content constant while varying representation (Figure 1). It enables systematic comparisons across modeling families, task conditions, and evaluation dimensions. Our contributions include:

- Controlled comparison of representations: we introduce a verbalization pipeline to convert structured tables into semi-structured forms, isolating the effects of representation while preserving semantics.
- Fine-grained diagnostic benchmark: we decompose task difficulty into four dimensions including table size, table joins, query complexity, and schema quality. This enables targeted analysis of method strengths and weaknesses.
- Comprehensive evaluation: we assess NL2SQL, LLM-based, and hybrid methods on BIRD (Li et al., 2023), MMQA (Wu et al., 2025a), and TableEval (Zhu et al., 2025) datasets, revealing trade-offs that inform both model design and practical deployment.

Our experiments show that representation is a major driver of performance. NL2SQL excels on structured inputs but drops 30–45% on semi-structured ones. LLMs are more stable (only 3.5% decline) but struggle with long or compositional queries. Hybrids fall by under 5% and are generally robust across conditions. These trade-offs intensify with larger tables, complex queries, and noisy schemas. Long tables reduce accuracy for all methods, though NL2SQL remains strong on long structured tables (62.9% accuracy). In contrast, hybrids outperform LLMs on long semi-structured tables, demonstrating better scalability. Multi-table reasoning and query complexity remain challenging across all methods. Schema quality proves especially influential: noisy schemas severely hurt NL2SQL, while verbalization often improves LLMs and hybrids by embedding schema cues in natural language.

Ultimately, no single paradigm excels across all conditions. Our findings establish representation as a central factor in Table QA, provide actionable insights for method selection, and motivate future hybrid systems to operate robustly across diverse real-world table formats. We will release our framework and benchmark to support ongoing research.

## 2 EXPLORING POTENTIAL FACTORS OF REASONING

Table QA is challenging not only due to the diversity of natural language queries but also due to how information is represented. A model's performance depends jointly on table representation (structured vs. semi-structured) and additional factors such as table size, schema quality, and query complexity. These challenges are salient in real-world settings, where tables are often large, noisy, and irregularly organized. We first contrast structured and semi-structured representations, then introduce four core dimensions that govern reasoning difficulty and ground our diagnostic evaluation.

## 2.1 TABLE REPRESENTATIONS: STRUCTURED VS. SEMI-STRUCTURED

**Structured tables.** Structured tables adhere to fixed schemas. Each column encodes a predefined attribute and each row conforms to the schema. This consistency enables efficient storage, precise querying, and symbolic approaches such as NL2SQL (Ramakrishnan & Gehrke, 2003). However, most NL2SQL datasets assume idealized schemas, overlooking real-world issues such as missing attributes or misaligned semantics. For instance, a query about the 'standing of Alice' becomes invalid if that information is embedded in free text rather than in a dedicated column (Figure 1).

**Semi-structured tables.** Semi-structured tables relax these constraints. Columns may be merged or repurposed (e.g., combining grade and standing), rows may summarize entities, and cells may contain long-form text, lists, or multimodal content. Web tables (Wang et al., 2024) and spreadsheets (Ma et al., 2024) exemplify this format. While they capture rich real-world information, their irregular structure complicates parsing, schema alignment, and reasoning.

## 2.2 CORE DIMENSIONS OF REASONING DIFFICULTY

Prior studies have shown that representational aspects such as table length (Chen et al., 2024), multi-table reasoning (Wu et al., 2025a), and query complexity (Zhu et al., 2025) substantially affect performance. Yet, these insights remain fragmented across benchmarks, and no framework has systematically examined their interaction with representation. We close this gap by analyzing four core dimensions: *table size*, *table joins*, *query complexity*, and *schema quality*. By varying these conditions under both structured and semi-structured representations, we disentangle their individual contributions and analyze robustness.

**Table size.** Large tables strain token windows and increase attention cost quadratically, leading to truncation, sparse supervision, and difficulty locating relevant rows Liu et al. (2023); Hsieh et al. (2024). LLMs also approximate numbers and struggle with exact comparisons across many values, causing errors in factual and financial QA Chen et al. (2019); Zhu et al. (2021). In this work, we categorize *short tables* as under 100 rows and *long tables* as over 100 rows. We focus on tables that still fit into the context windows of current LLMs. Extending the framework to ultra-long tables with hundreds or thousands of rows, which likely require chunking or retrieval, is an interesting direction for future work.

**Table joins.** Joins require aligning schemas and linking rows across multiple tables. This introduces challenges of ambiguous keys, irrelevant columns/rows and compositional reasoning. Semi-structured representations exacerbate these issues since explicit keys are often absent. We vary the number of tables per query and the presence of explicit key constraints to probe join difficulty.

**Query complexity.** Queries range from simple lookups to multi-step reasoning involving aggregation, filtering, comparisons, and arithmetic. Existing datasets often target narrow slices (e.g., SQL lookups (Zhong et al., 2017), logical forms over web tables (Pasupat & Liang, 2015), or discrete reasoning (Zhu et al., 2021)). It remains unclear how methods scale from lookups to multi-step reasoning, particularly across table representations.

**Schema quality.** Schema encodes column names, data types, keys, and relationships. While curated databases maintain consistent schemas, real-world tables often have missing headers, inconsistent formatting, and absent metadata. Such noise can severely affect symbolic methods.

Together, these factors define the major dimensions of Table QA difficulty. By systematically varying them under both structured and semi-structured settings, we disentangle their individual contributions and assess model robustness.

## 3 REPAIRTQA: FINE-GRAINED DIAGNOSTIC BENCHMARK

To systematically study the impact of table representations and disentangle the effects of the four factors outlined in Section 2, we construct REPAIRTQA, a fine-grained diagnostic benchmark. Our design follows two guiding principles: (1) **Representation pairing.** Comparing structured and semi-structured Table QA requires identical semantics; otherwise, differences may reflect content mismatches. We achieve this by transforming structured tables into semi-structured ones via a con-

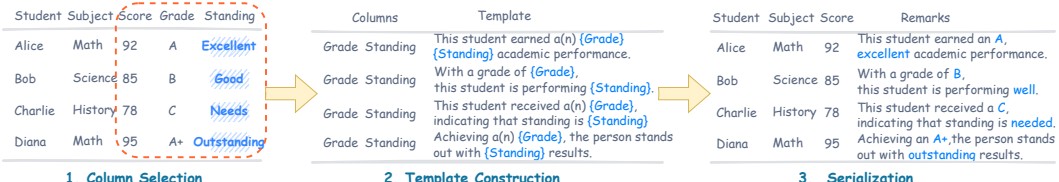

Figure 2: Verbalization pipeline for transforming structured tables into semi-structured representations while preserving semantics.

trolled verbalization strategy. (2) **Factor isolation.** We partition existing benchmarks into controlled subsets that vary along the core factors. This allows us to assess each factor's impact independently.

## 3.1 VERBALIZATION PIPELINE

Given a QA pair over a structured table, our verbalization pipeline produces a semi-structured variant of the same table. Each question is thus paired with two versions of its table: structured and semi-structured. The pipeline consists of three steps, illustrated in Figure 2:

1. **Column selection.** We first determine which columns to verbalize into free text. GPT-4o identifies suitable candidate columns, from which we sample *random* combinations to introduce diversity across instances. We experimented with three alternative selection strategies (Appendix B) and found comparable performance trends. We, hence, adopt the random strategy as the default.

2. **Template construction.** For the selected columns, we generate natural language templates conditioned on the table schema, into which column values are inserted to form descriptive sentences. To reduce annotation cost, GPT-4o produces candidate templates, which are lightly corrected for errors. For each column combination, we create five diverse templates to increase variation.

3. **Serialization.** Finally, templates are instantiated with table values, merging verbalized content into a single free-text column while removing the original structured columns. The resulting semi-structured table mirrors the semantics of the original but encodes them in a text-heavy, less rigid format. Both versions are retained for side-by-side evaluation.

## 3.2 DIAGNOSTIC SPLITS

To isolate the effects of representational and structural factors, we partition benchmarks into controlled subsets rather than treating them as monolithic. Each split targets one of four dimensions: table size, number of tables, query complexity, and schema quality. This design allows us to identify settings where different paradigms succeed or fail and provides a fine-grained view of robustness not captured by benchmark-level scores.

We construct diagnostic splits from three complementary benchmarks: 1) **BIRD** (Li et al., 2023): a large-scale, curated dataset with clean schemas and reliable evaluation protocols, representing the best-case setting for structured Table QA. 2) **MMQA** (Wu et al., 2025a): a benchmark explicitly designed for multi-table reasoning, where questions require retrieving and integrating evidence across multiple relational tables. 3) **TableEval** (Zhu et al., 2025): a collection of real-world web tables with noisy schemas, incomplete metadata, and inconsistent formatting, reflecting practical deployment challenges. Together, these datasets span clean to noisy conditions and single- to multi-table reasoning, providing a broad testbed for our diagnostic evaluation. Table 1 summarizes the resulting fine-grained diagnostic splits.

## 4 EXPERIMENTAL SETUP

We now describe our evaluation protocol for isolating representation effects, including research questions, baselines, and metrics.

Table 1: Diagnostic splits for factors: table length, no. of tables, query complexity, schema quality.

| Subset ID | Data Source | Table Joins | Length | Query Complexity | Schema | #Samples |
|---|---|---|---|---|---|---|
| S1 | BIRD | ✗ | short | lookup | clean | 95 |
| S2 | TableEval | ✗ | short | lookup | incomplete | 169 |
| S3 | BIRD | ✗ | short | compositional reasoning | clean | 435 |
| S4 | BIRD | ✗ | long | lookup | clean | 78 |
| S5 | BIRD | ✗ | long | compositional reasoning | clean | 258 |
| M1 | MMQA | ✓ | short | compositional reasoning | clean | 427 |
| M2 | MMQA | ✓ | long | compositional reasoning | clean | 166 |

## 4.1 RESEARCH QUESTIONS

We organize our comparison across method families around five key questions: **RQ1.** How does performance vary across paradigms when comparing structured and semi-structured inputs with identical content? **RQ2.** How do methods scale from short ($<100$ rows) to long ($\geq 100$ rows) tables? **RQ3.** How do multi-table alignment and join operations affect performance? **RQ4.** How does performance change from simple lookups to multi-step compositional queries (filters, aggregations, joins, comparisons)? **RQ5.** How robust are methods under noisy or incomplete schemas?

## 4.2 BASELINES AND SETUPS

We evaluate three method families by selecting representative methods from each family.

- **LLMs.** We use GPT-4o[1] (Hurst et al., 2024), Gemini-2.5-flash[2] (Anil et al., 2023), and Qwen3[3](Yang et al., 2025) as baselines for directly predicting the answer without any SQL execution. The detailed prompts are shown in Appendix A.
- **NL2SQL methods.** We include two approaches that generate SQL queries. a) **LLM-NL2SQL baseline**: a two-stage pipeline where an LLM generates SQL from the question and schema, executed via SQLite. We provide schema information (tables, columns, keys) and a small set of example values per column via value previews. The detailed prompts are shown in Appendix A. b) **XiYan** (Gao et al., 2024): a recent framework that improves SQL generation via a multi-generator ensemble design. It includes modules for schema linking via column/value retrieval, candidate generation with diverse LLM-based generators, and candidate selection/ranking before execution. We use GPT-4o as backbone across generators and rankers for fair comparison.
- **Hybrid methods.** We consider two hybrid approaches that combine SQL-based retrieval with LLM reasoning. a) **H-STAR** (Abhyankar et al., 2025): a hybrid method that first performs table extraction (relevant columns/rows via SQL and LLM filtering), then routes numeric/aggregation tasks to SQL and relational/descriptive reasoning to the LLM. This allows H-STAR to combine precise symbolic computation with flexible natural-language inference. b) **Weaver** (Khoja et al., 2025): executes a stepwise workflow assigning operations to SQL or LLM, feeding intermediate tables back into the plan for integrated reasoning. This tight integration allows it to combine the precision of SQL with the flexibility of LLMs, improving robustness on complex queries that require both symbolic computation and semantic interpretation.

## 4.3 EVALUATION

Previous work (Deng et al., 2022; Wu et al., 2025a; Zhu et al., 2021) has largely relied on traditional metrics such as Exact Match (EM) and Partial Match (PM). However, these metrics suffer from well-known limitations: they depend on strict string-level comparison with the gold answer. Consequently, they penalize semantically correct predictions due to paraphrasing, formatting, or minor rounding differences. To address these limitations, we adopt LLMs as judges, a strategy shown to correlate strongly with human evaluation in a variety of reasoning tasks (Maekawa et al., 2025). Concretely, GPT-4o is given both the gold answer and the model's prediction and prompted to determine semantic correctness. This evaluator tolerates surface variation while enforcing strict semantic equivalence. To assess the reliability of our LLM-based evaluator, we randomly sampled 100 eval-

---

[1] `gpt-4o-2024-11-20`
[2] `gemini-2.5-flash`
[3] `qwen3-235b-a22b-thinking-2507`

Table 2: Comparison of model paradigms on structured vs. semi-structured tables (RQ1). Weighted averages across diagnostic splits highlight the strong impact of input representation. **Bold** denotes the best, *italic* the second-best result within each block.

| Model | Structured Acc. (%) | Semi-structured Acc. (%) | Drop (%) |
|---|---|---|---|
| GPT-4o | 45.37 | 41.93 | 3.44 |
| Gemini-2.5-flash | 52.07 | 50.78 | 1.29 |
| Qwen3-235B | 38.20 | 36.70 | 1.50 |
| LLM-NL2SQL | *69.14* | 38.65 | *30.49* |
| XiYan | **69.55** | 24.08 | **45.47** |
| H-STAR | 49.48 | *47.14* | 2.34 |
| Weaver | 62.19 | **57.70** | 4.49 |

uation cases across all datasets and model families. Each case was independently annotated by a human and compared to the LLM's judgment. We observed a 96% agreement rate, demonstrating that the LLM-based evaluation is sufficiently stable for large-scale benchmarking. Unless stated otherwise, all results reported follow this evaluation protocol. We provide a Chi-Square significance analysis in Appendix E to rigorously evaluate performance differences across data splits.

# 5  RESULTS AND ANALYSIS

## 5.1  RQ1: MODEL PERFORMANCE ACROSS PARADIGMS

We first examine how model performance varies across paradigms when comparing structured and semi-structured inputs with identical content, using weighted averages over all diagnostic splits.

**NL2SQL models achieve the highest accuracy on structured tables, while hybrid models perform best on semi-structured tables.** Table 2 shows that input representation has a substantial and consistent effect on performance, with semi-structured inputs posing greater challenges across the methods. On structured inputs, SQL-based methods outperform both hybrids and LLMs, reflecting their reliance on clean, explicit schemas. On semi-structured inputs, this ranking reverses. Hybrid models achieve the highest absolute accuracy on semi-structured tables, while LLMs exhibit relatively modest performance drops. SQL methods suffer dramatic drops since irregular or implicit structures disrupt symbolic execution.

We also examine the robustness of these trends across model scales. For instance, Gemini-2.5-Pro shows the same structured vs. semi-structured performance gap as GPT-4o and other baselines, indicating that sensitivity to representation persists even in state-of-the-art LLMs (full results in Appendix D).

**LLMs are the most robust to representation changes, due to their training on free-text, though they do not attain peak accuracy in either setting.** These trends highlight the central role of representation in shaping model performance.

## 5.2  RQ2: IMPACT OF TABLE SIZE

We next examine how table size affects model performance across paradigms. To isolate the effect of table length, we compare short tables S1 and S3 against long tables S4 and S5 (see Table 1).

**NL2SQL pipelines scale best on structured data but collapse on semi-structured tables.** Table 3 indicates that table size has a substantial effect on accuracy across all families. Performance declines monotonically as tables grow. The rate of degradation, however, depends on the paradigm. NL2SQL methods perform best on short, structured tables and remain competitive on longer structured ones, leveraging execution engines that scale efficiently. However, semi-structured inputs expose their reliance on clean schemas. Accuracy drops sharply at scale (e.g., XiYan drops to 25.0%).

**LLMs are highly sensitive to table length, whereas hybrids offer a balanced trade-off.** LLMs perform reasonably on short tables but degrade substantially on long tables. For example, GPT-4o falls to 28.9% on structured and 27.1% on semi-structured data; with Gemini falling to <15%. Hybrid models, while less competitive on short structured tables, maintain relative robustness across

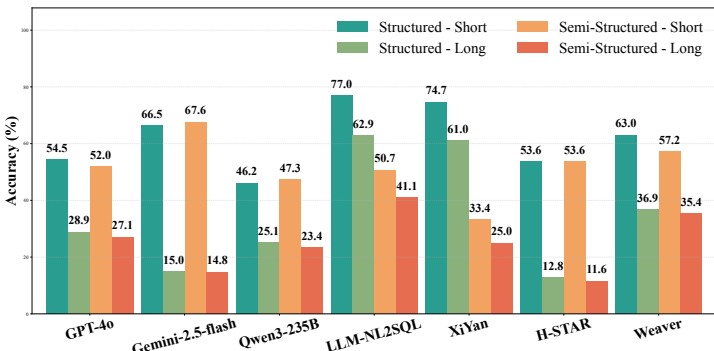

Figure 3: Short tables vs. long tables (RQ2). All models struggle on long tables. LLMs are most sensitive, NL2SQL excels on short structured tables, and hybrids remain relatively stable.

table sizes, particularly on semi-structured data. For example, Weaver achieves 57.2% on short semi-structured tables and 35.4% on long ones, while H-STAR drops to <13% on long tables regardless of representation.

We also examine the robustness of these trends across table width (Appendix C). We found that reasoning difficulty grows with table width: wide tables (>20 columns) consistently challenge all model families as it makes schema linking harder and context retrieval noisier.

## 5.3  RQ3: IMPACT OF TABLE JOINS

To assess the effect of multi-table reasoning, we compare single-table subsets (S1–S5) with multi-table subsets (M1–M2).

**NL2SQL models benefit from executable joins on structured tables but lose this advantage when schema signals are weak.** Table 4 indicates that multi-table reasoning has a significant impact on performance, with effects varying by paradigm. On structured multi-table inputs, NL2SQL models leverage joins to integrate information across tables. For instance, LLM-NL2SQL improves from 71.5% (single-table) to 82.3% (multi-table), outperforming GPT-4o by over 35 points. However, this advantage disappears under semi-structured conditions, where implicit structure hinders grounding, resulting in sharp performance drops.

**LLMs are largely insensitive to join structure.** They show minimal differences between single- and multi-table inputs, reflecting limited ability to exploit relational dependencies. Gemini exhibits modest improvements on multi-table structured inputs, but gains are small relative to SQL-based methods. Hybrid models maintain relatively stable performance on structured multi-table inputs but currently offer limited support for multi-table reasoning under semi-structured conditions, constraining their effectiveness in realistic deployments.

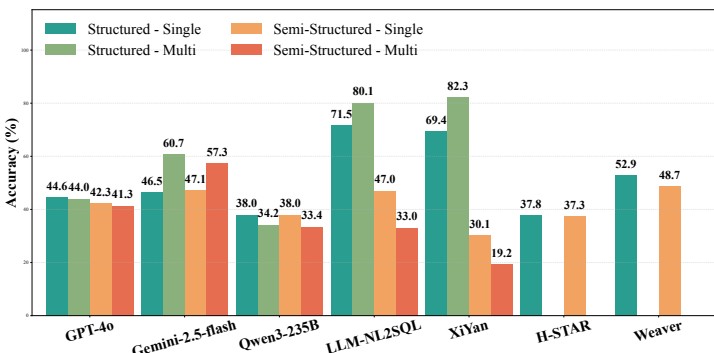

Figure 4: Single- vs. multi-table (RQ3). NL2SQL benefits from structured joins but fails on semi-structured tables. LLMs are largely insensitive to joins, while hybrids lack multi-table support.

## 5.4 RQ4: IMPACT OF QUERY COMPLEXITY

We assess how query complexity affects performance by comparing *lookup* (subsets S1, S4) and *compositional* (subsets S3, S5) queries. These subsets target single tables. To isolate impact of query complexity we exclude M1, M2 since they focus on multi-table settings.

**NL2SQL models excel on structured queries but collapse under semi-structured conditions.** Figure 5 shows that NL2SQL models achieve high accuracy on structured compositional queries, confirming the advantage of executable SQL for complex reasoning. However, their performance drops sharply under semi-structured conditions, reflecting brittleness to representational noise.

**LLMs perform well on lookup queries but struggle with multi-hop reasoning.** They achieve 70% accuracy on simple lookups but degrade significantly on compositional queries, highlighting limitations in symbolic and multi-step reasoning without execution support.

**Hybrid models occupy the middle ground.** They experience moderate drops from lookup to compositional queries, balancing flexibility and precision. In particular, Weaver sometimes outperforms structured inputs on semi-structured lookup queries, suggesting that natural-language verbalizations can better align with LLM pretraining and improve reasoning.

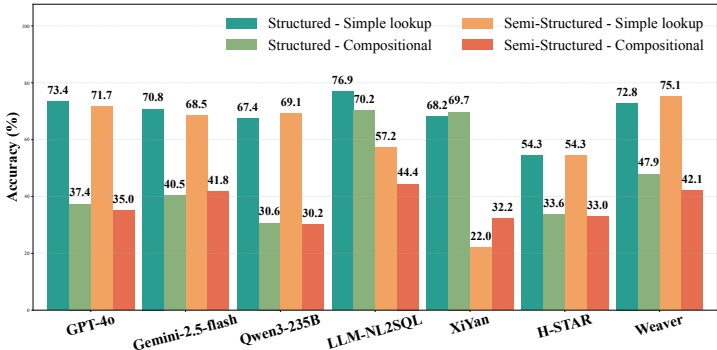

Figure 5: Lookup vs. compositional queries (RQ4). Accuracy drops across all models on multi-hop queries. NL2SQL excels on structured but fails on semi-structured inputs. LLMs and hybrids also degrade, though hybrids often benefit from semi-structured lookup queries.

## 5.5 RQ5: IMPACT OF SCHEMA QUALITY

We assess the effect of schema quality by comparing clean-schema tables (S1) with incomplete-schema tables (S2). We focus on lookup queries to isolate schema effects.

**NL2SQL models are highly sensitive to schema quality, while LLMs and hybrids are more robust.** Figure 6 shows that NL2SQL performance drops under incomplete schemas, reflecting their dependence on well-defined metadata. In contrast, LLMs maintain relatively high accuracy on clean schemas and degrade only moderately on incomplete ones, leveraging surface patterns even in noisy tables. Hybrid models are more resilient to schema noise. H-STAR mitigates irregularities via row/column pruning and alternative table views. Weaver remains stable by automatically renaming columns during SQL execution, reducing schema mismatches.

## 5.6 CASE STUDY

We now present illustrative examples that capture the main failure modes when shifting from structured to semi-structured representations. Figure 7 shows a representative case from the "supplier" table, where the query asks for the top ten suppliers by account balance. On the structured version, NL2SQL methods (LLM-NL2SQL and XiYan) execute correctly by leveraging explicit fields such as "s_suppkey" and "s_acctbal". On the semi-structured version, however, these attributes are embedded into free-text descriptions. The same models either output incorrect results or raise execution errors due to missing column references, illustrating the brittleness of NL2SQL pipelines under representational shifts. A similar failure is observed for the hybrid method Weaver, which also relies on

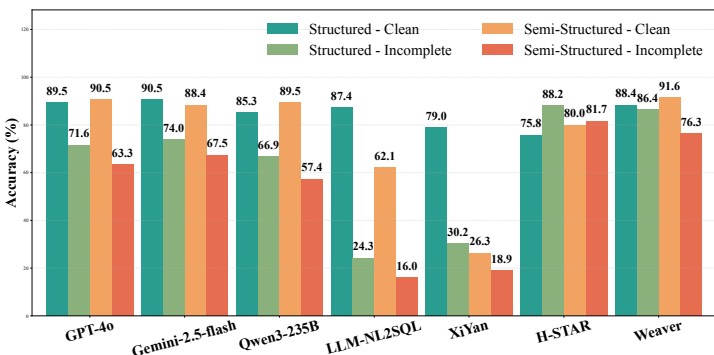

Figure 6: Clean-schema vs. incomplete-schema with lookup queries (RQ5). NL2SQL degrades under schema noise, while LLMs and hybrids remain robust.

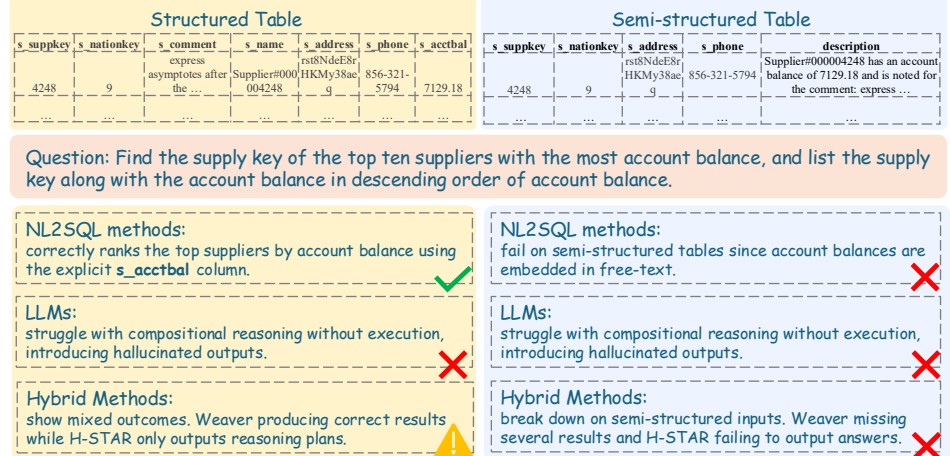

Figure 7: An illustrative case from the RQ1 experiments. A check mark or cross indicates that all models in the series answered correctly or incorrectly, respectively, while the warning symbol denotes mixed outcomes. Example also compares structured and semi-structured inputs.

SQL execution and cannot recover when key attributes are verbalized into text. This example highlights the challenges that semi-structured representations pose for models that depend on explicit schema linking and executable queries. We also provide additional case studies in Appendix G that highlight scenarios where different model families diverge in behavior.

## 6  DISCUSSION & FUTURE WORK

We summarize the optimal method selection strategy as a decision tree in Figure 8. On the structured data, schema quality is the key discriminator. On clean, descriptive schemas, NL2SQL systems perform best. However, when schemas are incomplete or ambiguous, hybrid pipelines are preferable since their retrieval, normalization, and schema–query alignment steps compensate for missing metadata. On the *semi-structured* data, the first branch is table size. For long tables, NL2SQL becomes competitive once a stable schema mapping is established, whereas for short tables, query complexity dominates. Simple lookups favor LLM-only reasoning, while compositional or multistep queries benefit from hybrid pipelines.

A deeper theoretical understanding of representation sensitivity is an important avenue for future work. Our focus here is on establishing a controlled benchmark and documenting empirical trends. Fully explaining the underlying mechanisms will require tools such as architectural ablations, tar-

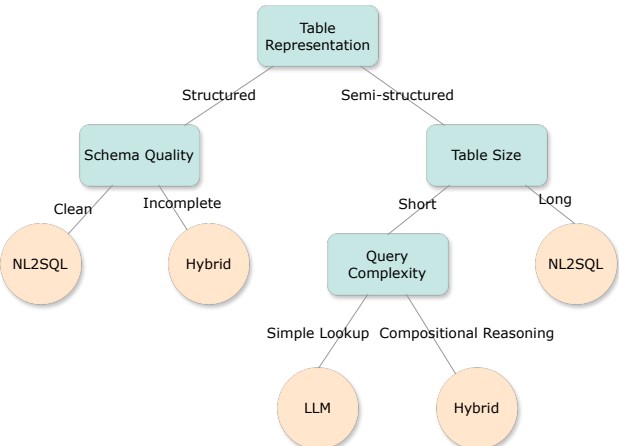

Figure 8: Practical decision tree for method selection. NL2SQL excels in structured, clean tables, while Hybrid methods offer superior robustness for semi-structured, noisy, or large-scale contexts. LLMs serve as cost-effective baselines for simple lookups.

geted probing of intermediate representations, and instrumentation of model-internal decision pathways. Developing such analyses on top of our benchmark is a promising next step.

## 7 RELATED WORK

Answering questions over tables requires models to reason over information that can appear in both structured database-style formats (Li et al., 2023; Lei et al., 2025) and semi-structured natural-language-like representations (Chen et al., 2021; 2020). Approaches to this problem can be broadly grouped into three paradigms, including translating questions into SQL queries (Wang et al., 2025), directly reasoning over table representations with large language models (Wu et al., 2025b; Zhang et al., 2025b), and hybrid methods (Khoja et al., 2025; Abhyankar et al., 2025) that combine symbolic and neural reasoning. Existing benchmarks (Zhong et al., 2017; Zhu et al., 2025; Wu et al., 2025a) for table QA typically categorize data by query type or task category, but pay little attention to representation factors such as table size, table joins, or schema quality together. Crucially, no prior work has isolated the effect of semi-structured representations under the same information conditions. Meanwhile, most analytical studies (Ashury-Tahan et al., 2025; Singha et al., 2023; Sui et al., 2024) evaluate a single family of methods in isolation, without systematically comparing how different paradigms respond to representation shifts. Our work fills these gaps by conducting a controlled, same-information study of structured and semi-structured inputs, analyzing their interaction with key reasoning factors across multiple method families.

## 8 CONCLUSION

We performed a controlled, same-information comparison of structured and semi-structured table representations across LLM, NL2SQL, and hybrid paradigms using a fine-grained diagnostic dataset constructed via an information-preserving verbalization pipeline. Our findings confirm that representation has an first-order effect. NL2SQL attains peak accuracy on structured inputs yet is brittle under semi-structured or noisy schemas. LLMs are comparatively robust across formats but trade off peak accuracy. Hybrids mediate this trade-off and often lead under semi-structured inputs. Task factors consistently influence these trends. Long tables degrade all methods (most for LLMs), explicit joins benefit NL2SQL on structured data, multi-hop queries challenge all paradigms (especially NL2SQL on semi-structured inputs), and schema incompleteness severely impacts SQL-based systems. Overall, our results highlight the need for representation-aware benchmarks and systems, and for matching method families to data conditions in deployment.

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

# A  PROMPTS

---
**Prompt for Random Column Selection**

**Prompt:**  You are given a list of column names from a database table, along with a sample value for each column. Your task consists of two steps:

Step 1: Column Suitability Classification
Label each column as either:
- "YES": if the column is descriptive and human-interpretable (e.g., name, title, category, price, quantity, location, status).
- "NO": if it is a technical field or metadata column (e.g., id, uuid, product_id, created_at, URL) that is unlikely to appear in a natural-language description.

Step 2: Column Combination Generation
From the columns labeled "YES", generate a small number of diverse and meaningful combinations (ideally 2–5). Each combination should:
- Contain only columns labeled "YES"
- Include several columns (based on availability, usually should more than two)
- Avoid repeating the same column in a single combination
- Be suitable for use in a natural-language sentence such as: "{product_name} is sold by {supplier_name} for {price}."
If only one or two YES columns exist, just generate 1 or 2 simple combinations. Avoid bad combinations like: ["ATT_CLASS", "ATT_CLASS"] ["ATT_CLASS_ID"] if it was labeled "NO"

Input:
Database: db Table: table
Columns with sample values: cols

Example:
Columns = [product_name, price, category, product_id, created_at, stock_quantity, supplier_name] Example Output: [ ["product_name", "price", "category"], ["product_name", "supplier_name"], ["product_name", "stock_quantity", "price"] ]

Your Output Format:
"'json { "column_classification": { "column_name_1": "YES", "column_name_2": "NO" }, "combinations": [ ... ] } "'

---
**Prompt for Random Column Selection (One Column)**

**Prompt:**  You are given a list of column names from a database table, along with a sample value for each column. Your task consists of two steps:

Step 1: Column Suitability Classification
Label each column as either:
- "YES": if the column is descriptive and human-interpretable (e.g., name, title, category, price, quantity, location, status).
- "NO": if it is a technical field or metadata column (e.g., id, uuid, product_id, created_at, URL) that is unlikely to appear in a natural-language description.

Step 2: Column Combination Generation
From the columns labeled "YES", generate a small number of diverse and meaningful combinations (ideally 2–5). Each combination should:
- Contain only columns labeled "YES"
- Include only one column
- Avoid repeating the same column in a single combination

---
16

- Be suitable for use in a natural-language sentence such as: "{product_name} is sold by {supplier_name} for {price}."
If only one or two YES columns exist, just generate 1 or 2 simple combinations. Avoid bad combinations like: ["ATT_CLASS", "ATT_CLASS"] ["ATT_CLASS_ID"] if it was labeled "NO"

Input:
Database: db Table: table
Columns with sample values: cols

Example:
Columns = [product_name, price, category, product_id, created_at, stock_quantity, supplier_name] Example Output: [ ["product_name", "price", "category"], ["product_name", "supplier_name"], ["product_name", "stock_quantity", "price"] ]

Your Output Format:
"'json { "column_classification": { "column_name_1": "YES", "column_name_2": "NO" }, "combinations": [ ... ] } "'

---

**Prompt for Random Column Selection (Three Column)**

**Prompt:** You are given a list of column names from a database table, along with a sample value for each column. Your task consists of two steps:

Step 1: Column Suitability Classification
Label each column as either:
- "YES": if the column is descriptive and human-interpretable (e.g., name, title, category, price, quantity, location, status).
- "NO": if it is a technical field or metadata column (e.g., id, uuid, product_id, created_at, URL) that is unlikely to appear in a natural-language description.

Step 2: Column Combination Generation
From the columns labeled "YES", generate a small number of diverse and meaningful combinations (ideally 2–5). Each combination should:
- Contain only columns labeled "YES"
- Include exactly three columns
- Avoid repeating the same column in a single combination
- Be suitable for use in a natural-language sentence such as: "{product_name} is sold by {supplier_name} for {price}."
If only one or two YES columns exist, just generate 1 or 2 simple combinations. Avoid bad combinations like: ["ATT_CLASS", "ATT_CLASS"] ["ATT_CLASS_ID"] if it was labeled "NO"

Input:
Database: db Table: table
Columns with sample values: cols

Example:
Columns = [product_name, price, category, product_id, created_at, stock_quantity, supplier_name] Example Output: [ ["product_name", "price", "category"], ["product_name", "supplier_name"], ["product_name", "stock_quantity", "price"] ]

Your Output Format:
"'json { "column_classification": { "column_name_1": "YES", "column_name_2": "NO" }, "combinations": [ ... ] } "'

### Prompt for Random Column Selection (Six Column)

**Prompt:** You are given a list of column names from a database table, along with a sample value for each column. Your task consists of two steps:

Step 1: Column Suitability Classification
Label each column as either:
- "YES": if the column is descriptive and human-interpretable (e.g., name, title, category, price, quantity, location, status).
- "NO": if it is a technical field or metadata column (e.g., id, uuid, product_id, created_at, URL) that is unlikely to appear in a natural-language description.

Step 2: Column Combination Generation
From the columns labeled "YES", generate a small number of diverse and meaningful combinations (ideally 2–5). Each combination should:
- Contain only columns labeled "YES"
- Include exactly six columns
- Avoid repeating the same column in a single combination
- Be suitable for use in a natural-language sentence such as: "{product_name} is sold by {supplier_name} for {price}."
If only one or two YES columns exist, just generate 1 or 2 simple combinations. Avoid bad combinations like: ["ATT_CLASS", "ATT_CLASS"] ["ATT_CLASS_ID"] if it was labeled "NO"

Input:
Database: db Table: table
Columns with sample values: cols

Example:
Columns = [product_name, price, category, product_id, created_at, stock_quantity, supplier_name] Example Output: [ ["product_name", "price", "category"], ["product_name", "supplier_name"], ["product_name", "stock_quantity", "price"] ]

Your Output Format:
"'json { "column_classification": { "column_name_1": "YES", "column_name_2": "NO" }, "combinations": [ ... ] } "'

### Prompt for Template Generation

**Prompt:** You are writing five distinct fluent English sentence templates that verbalize one row of a database table.
Database: db Table: table
Required columns (each sentence must use every one exactly once as placeholders): sel_cols
Sample values for these columns (for context): example_block
Guidelines 1. **Produce exactly 5 sentences.** Each sentence must include every required column once and only once, wrapped as column_name. 2. You may reorder the placeholders naturally; no extra columns should be added. 3. Keep each sentence concise, grammatically correct, and natural to a human reader. 4. The column values are descriptive and human-interpretable (names, dates, quantities, locations, etc.). Avoid metadata fields like IDs or flags. 5. Return a single valid JSON object in the exact format below—no Markdown, no commentary:
"'json { "template1": "Sentence 1 with {col1}, {col2}, ...", "template2": "Sentence 2 with {col1}, {col2}, ...", "template3": "Sentence 3 with {col1}, {col2}, ...", "template4": "Sentence 4 with {col1}, {col2}, ...", "template5": "Sentence 5 with {col1}, {col2}, ..." }
Example for reference Columns = [product_name, price, category] Sample values = product_name: "iPhone 15", price: "899 USD", category: "Electronics" Expected output: "'json { "template1": "product_name in the category category is priced at {price}.", "template2":

"The {category} item {product_name} costs {price}.", "template3": "Retailing for {price}, the {product_name} belongs to {category}.", "template4": "{product_name}—a {category} product—carries a price tag of price.", "template5": "Price for the {category} product {product_name} is price." }

---

**Prompt for LLM inference**

**Prompt:** You are an expert at table question answering. You need to extract answers based on the following information:
[Tables]
{tables_md}
[Question] {question}
Please return your answer in JSON format only, with no explanation, following this structure:
{"Answer": "[your answer]"}

---

**Prompt for NL2SQL Baseline**

**Prompt:** You are a careful NL2SQL expert for SQLite. Rules: - Return a single SQL query that answers the question.
- Use ONLY provided tables/columns.
- Prefer explicit JOINs with ON.
- Use SQLite syntax.
- Output ONLY SQL inside one fenced block:
```sql SELECT ... ```
SQLite schema & preview: {schema_md}
Question: {question}

---

**Prompt for Evaluation**

**Prompt:** You are an expert in evaluating question answering results.
[Ground-truth Answer] {ground_truth}
[Model Prediction] {prediction}
Is the prediction semantically equivalent to the ground-truth answer?
Please respond with only one word: "CORRECT" or "INCORRECT".

## B  EFFECT OF COLUMN SELECTION SETTINGS

We test whether our conclusions are sensitive to how content is chosen for verbalization. On the S3 split, we keep the questions and gold answers fixed and vary *only* the selection setting. We consider four settings: *query-related*, where only the columns directly referenced in the question are verbalized; *non–query-related*, where only the columns not referenced in the question are verbalized; *random*, where GPT-4o is prompted to randomly select a subset of columns to verbalize; and *full*, where all columns in the table are verbalized into free-text. For the *full* strategy, directly verbalizing all rows would collapse the table into a single unstructured text block. To avoid this, we randomly verbalize only an $\alpha\%$ subset of rows, while keeping the remaining rows in their original structured form. The verbalized rows are then dropped, and the resulting semi-structured table contains both the structured part and the generated free-text column. We report $\alpha = 50$ and $\alpha = 10$ below.

For tables with fewer columns than the target verbalization count, we discard those cases; all results are computed on the intersection of examples available under every selection setting to ensure strict comparability.

**Findings.** (1) Across selection settings, GPT-4o accuracy spans 42.20%–46.45%, indicating low sensitivity to the specific choice. (2) *Column verbalization* exhibits a clear downward trend as the number of verbalized columns increases, with the unconstrained/random setting producing the

Table 3: Effect of verbalization strategy on the S3. We vary only the selection policy while keeping questions and gold answers fixed.

| Settings | Acc (%) |
|---|---|
| *Column verbalization* | |
|     Random (1 column) | 44.68 |
|     Random (3 columns) | 43.97 |
|     Random (6 columns) | 42.20 |
|     Random (unconstrained count) | 42.20 |
| *Query-aware* | |
|     Query-Related | *46.45* |
|     Non–Query-Related | 43.26 |
| *Full verbalization* | |
|     $\alpha=50\%$ rows | 43.97 |
|     $\alpha=10\%$ rows | 42.91 |

lowest score (42.20%). (3) *Query-aware* selection is the only variant that yields a consistent gain: verbalizing only the query-relevant columns gives the best result (46.45%), while excluding them hurts (43.26%). (4) *Full verbalization* with $\alpha = 50\%$ or $10\%$ behaves similarly to random column verbalization and offers no additional advantage. Given the comparable accuracy across all settings, we adopt *random column selection* as the default to maximize coverage and comparability in the main experiments.

## C  EFFECT OF TABLE WIDTH

While our primary analysis differentiates tables by row count, the number of columns (table width) is another important complexity dimension, especially in domains like medical or financial research where high-dimensional data is common. To explore this, we further divide the BIRD diagnostic splits (S1, S3, S4, S5) into three categories based on column count: *Narrow* (1–5 columns), *Medium* (6–20 columns), and *Wide* (>20 columns). Table 4 summarizes the distribution of instances across these categories. Table 5 reports the performance across different paradigms. Broadly, we observe an inverse correlation between table width and accuracy across most methods.

On structured tables, performance degrades consistently as width increases. For instance, GPT-4o drops from 52.75% on narrow tables to 27.91% on wide tables. SQL-based methods demonstrate superior resilience on wide tables, likely because symbolic SQL queries can effectively select relevant columns. On semi-structured tables, the challenge is amplified for LLMs. Models like GPT-4o and H-STAR suffer significant drops on wide tables. Interestingly, NL2SQL maintains consistent performance, further validating that symbolic execution provides a stable fallback when the input representation becomes noisy or high-dimensional.

Table 4: Distribution of table width buckets across BIRD-based diagnostic splits.

| Width Bucket | Column Range | # Instances |
|---|---|---|
| Narrow | $1-5$ | 455 |
| Medium | $6-20$ | 329 |
| Wide | $>20$ | 89 |

## D  ADDITIONAL RESULTS ON A LARGE INSTRUCTION-TUNED LLM: GEMINI-2.5-PRO

To validate the generalizability of our findings, we evaluated Gemini-2.5-Pro using the same setup as in Section 3.2. The results are summarized in Table 6. While Gemini-2.5-Pro demonstrates strong overall capabilities, achieving high accuracy on simpler structured queries (e.g., 92.63% on S1 Structured), it shows the same sensitivity to representation as GPT-4o and other baselines. We

Table 5: Performance comparison across table width buckets. **Bold** indicates the best performance within each setting and bucket.

| Model | Structured Accuracy (%) | | | Semi-Structured Accuracy (%) | | |
|---|---|---|---|---|---|---|
| | Narrow | Medium | Wide | Narrow | Medium | Wide |
| GPT-4o | 52.75 | 37.80 | 27.91 | 48.79 | 37.92 | 25.58 |
| Gemini-2.5-flash | 51.87 | 43.47 | 31.46 | 54.29 | 41.34 | 31.46 |
| Qwen3-235B | 42.64 | 34.35 | 28.09 | 43.96 | 31.91 | 30.34 |
| LLM-NL2SQL | **76.48** | **65.55** | **57.30** | 48.79 | 43.90 | **47.19** |
| XiYan | 75.16 | 64.94 | 56.18 | 24.18 | 35.06 | 43.82 |
| H-STAR | 43.30 | 32.01 | 25.84 | 44.84 | 32.52 | 21.35 |
| Weaver | 54.41 | 44.07 | 38.20 | **57.71** | **51.06** | 33.71 |

found a consistent accuracy drop when moving from structured to semi-structured tables across all difficulty levels. For instance, on the most complex subset (S5), the accuracy significantly declines from 41.31% (Structured) to 33.98% (Semi-Structured).

Table 6: Performance of Gemini-2.5-Pro across all splits (S1-S5 and M1-M2). The model consistently shows performance degradation on semi-structured inputs.

| Split | Structured Acc. (%) | Semi-Structured Acc. (%) | $\Delta$ (pp) |
|---|---|---|---|
| S1 | 92.63 | 88.42 | -4.21 |
| S2 | 81.07 | 73.37 | -7.70 |
| S3 | 76.59 | 73.86 | -2.73 |
| S4 | 60.76 | 59.49 | -1.27 |
| S5 | 41.31 | 33.98 | -7.33 |
| M1 | 82.44 | 78.45 | -3.99 |
| M2 | 35.54 | 29.52 | -6.02 |

## E  STATISTICAL SIGNIFICANCE ANALYSIS

To rigorously evaluate the performance differences across various data splits, we conduct Chi-Square tests for each comparison. We report significance levels at $p < 0.05$ (*), $p < 0.01$ (**), and $p < 0.001$ (***).

### E.1  TABLE LENGTH: SHORT VS. LONG

Table 7 presents the performance comparison between short and long tables. All methods show statistically significant performance degradation when processing long tables ($p < 0.001$), indicating that table length is a universal challenge. Gemini-2.5-flash exhibits the largest drop (52.17%), while LLM-NL2SQL and XiYan demonstrate relatively better robustness with drops of 11.87% and 11.06%, respectively.

Table 7: Performance comparison by table length (Short vs. Long).

| Model | Short (%) | Long (%) | Drop (%) | p-value | Sig. |
|---|---|---|---|---|---|
| GPT-4o | 53.26 | 27.98 | 25.29 | 3.38e-21 | *** |
| Gemini-2.5-flash | 67.06 | 14.88 | 52.17 | 1.37e-83 | *** |
| Qwen3-235B | 46.72 | 24.24 | 22.48 | 2.32e-17 | *** |
| LLM-NL2SQL | 63.86 | 51.99 | 11.87 | 8.17e-06 | *** |
| XiYan | 54.06 | 43.00 | 11.06 | 4.75e-05 | *** |
| H-STAR | 53.59 | 12.19 | 41.40 | 3.70e-55 | *** |
| Weaver | 60.09 | 36.16 | 23.94 | 8.35e-19 | *** |

## E.2 Number of Tables: Single vs. Multi

Table 8 shows the performance comparison between single-table and multi-table queries. Interestingly, not all methods are significantly affected by the number of tables. GPT-4o ($p = 0.789$), Qwen3-235B ($p = 0.097$), LLM-NL2SQL ($p = 0.306$), and XiYan ($p = 0.736$) show no statistically significant difference, suggesting robustness to table joins. In contrast, Gemini-2.5-flash and H-STAR perform significantly better on multi-table queries, while Weaver shows significant degradation.

Table 8: Performance comparison by number of tables (Single vs. Multi).

| Model | Single (%) | Multi (%) | $\Delta$ (%) | p-value | Sig. |
|---|---|---|---|---|---|
| GPT-4o | 43.45 | 42.66 | 0.79 | 7.89e-01 | |
| Gemini-2.5-flash | 46.81 | 59.03 | -12.22 | 2.47e-06 | *** |
| Qwen3-235B | 38.00 | 33.81 | 4.19 | 9.71e-02 | |
| LLM-NL2SQL | 59.25 | 56.57 | 2.68 | 3.06e-01 | |
| XiYan | 49.77 | 50.76 | -0.98 | 7.36e-01 | |
| H-STAR | 37.75 | 57.34 | -19.59 | 3.38e-14 | *** |
| Weaver | 52.89 | 33.39 | 19.50 | 4.97e-14 | *** |

## E.3 Query Complexity: Lookup vs. Compositional

Table 9 presents the performance comparison between simple lookup queries and compositional reasoning queries. Most methods show significant performance degradation on compositional queries ($p < 0.001$), with Qwen3-235B exhibiting the largest drop (37.79%). Notably, XiYan is the only method that shows no statistically significant difference ($p = 0.060$), and even performs slightly better on compositional queries, suggesting its reasoning capability. LLM-NL2SQL also demonstrates relatively strong compositional reasoning with only a 9.75% drop.

Table 9: Performance comparison by query complexity (Lookup vs. Compositional).

| Model | Lookup (%) | Comp. (%) | Drop (%) | p-value | Sig. |
|---|---|---|---|---|---|
| GPT-4o | 72.55 | 36.19 | 36.36 | 4.36e-33 | *** |
| Gemini-2.5-flash | 69.68 | 41.11 | 28.58 | 1.29e-20 | *** |
| Qwen3-235B | 68.24 | 30.45 | 37.79 | 9.77e-37 | *** |
| LLM-NL2SQL | 67.05 | 57.30 | 9.75 | 1.54e-03 | ** |
| XiYan | 45.09 | 50.94 | -5.85 | 6.01e-02 | |
| H-STAR | 54.34 | 33.33 | 21.01 | 1.60e-12 | *** |
| Weaver | 73.98 | 45.02 | 28.96 | 3.10e-21 | *** |

## E.4 Schema Quality: Clean vs. Noisy

Table 10 shows the performance comparison between clean and noisy (incomplete) schemas. All methods show statistically significant performance differences. LLM-NL2SQL shows the most severe degradation (54.62%), indicating high sensitivity to schema quality. Interestingly, H-STAR is the only method that performs better on noisy schemas ($p = 0.035$), suggesting potential robustness to schema imperfections. Thanks to table preprocessing, Weaver maintains relatively stable performance with only an 8.64% drop.

These results show that all the analysis supports our findings in the Section 5.

## F Fine-grained Results

Beyond the overall trends summarized in Section 5, we highlight several noteworthy findings from the fine-grained results in Table 11:

Table 10: Performance comparison by schema quality (Clean vs. Noisy).

| Model | Clean (%) | Noisy (%) | Drop (%) | p-value | Sig. |
|---|---|---|---|---|---|
| GPT-4o | 90.00 | 67.45 | 22.55 | 9.75e-17 | *** |
| Gemini-2.5-flash | 89.47 | 70.71 | 18.77 | 3.61e-12 | *** |
| Qwen3-235B | 87.37 | 62.13 | 25.24 | 1.37e-17 | *** |
| LLM-NL2SQL | 74.74 | 20.12 | 54.62 | 2.74e-47 | *** |
| XiYan | 52.64 | 24.56 | 28.08 | 1.40e-11 | *** |
| H-STAR | 77.90 | 84.91 | -7.02 | 3.52e-02 | * |
| Weaver | 90.00 | 81.36 | 8.64 | 6.97e-04 | *** |

- **NL2SQL models remain advantageous at scale.** Even with semi-structured inputs, NL2SQL sometimes outperforms LLMs and hybrids on long tables (e.g., LLM-NL2SQL 41.1% vs. Weaver 35.4% and GPT-4o 27.1% on S4), showing that SQL execution can still offset representation mismatch when contexts are large.

- **NL2SQL excels with joins.** On multi-table structured inputs, LLM-NL2SQL rises from 71.5% (single-table) to 82.3%, exceeding GPT-4o by more than 35 points. This highlights that executable joins are a unique strength of NL2SQL, whereas GPT-4o is largely insensitive to join structure.

- **Verbalization can improve simple lookup queries.** For simple lookup queries with clean schemas, aligning headers and content with natural language boosts hybrid models (e.g., Weaver $\sim$ +3 points on S1), demonstrating that verbalization not only enables semi-structured evaluation but can also yield accuracy gains.

# G ADDITIONAL CASE STUDIES

We present additional case studies to further illustrate the distinctive behaviors of different methods.

## G.1 MODEL BEHAVIOR ON LONG TABLES

Figure 9 shows a long user profile table with hundreds of rows and a query *"What is the id of the youngest user?"*. LLM-NL2SQL correctly answers the query on the structured version, which exposes the Age column and allows SQL execution to identify the minimum value (id = 7828). In contrast, LLMs fail to reliably scan the full column and return arbitrary large ids. Hybrid methods behave similarly. Weaver returns an arbitrary large id, while H-STAR concludes that the task is impossible because of numerous `NaN` values. This example illustrates that while NL2SQL with execution remains competitive on structured long tables, both LLMs and current hybrid methods are highly sensitive to table length and may produce spurious results when the input context becomes too large.

## G.2 MODEL BEHAVIOR ON INCOMPLETE SCHEMAS

Figure 10 presents two representative examples illustrating the impact of incomplete schemas on model performance. In the first example (left, RoboTaxi fare table), the Rest Day cell for 17:00–20:00 is left blank, although by business rules it should inherit the weekday rate (7 RMB). In this case, LLMs succeed by leveraging contextual cues to infer the missing value, while NL2SQL and hybrid pipelines fail due to their inability to capture implicit inheritance. The second example (right, biomedical citation table) involves a clinical reference for Alectinib that is embedded only within a free-text description column. In this case, NL2SQL methods return empty outputs, and LLMs hallucinate incorrect citations. In contrast, both hybrid methods (H-STAR and Weaver) successfully locate and extract the correct reference, demonstrating their ability to combine schema-guided retrieval with flexible reasoning over unstructured content. These examples illustrate that while NL2SQL pipelines are highly sensitive to missing schema entries, LLMs can sometimes compensate via contextual reasoning, and hybrid approaches offer a middle ground capable of handling certain forms of schema incompleteness when textual cues are present.

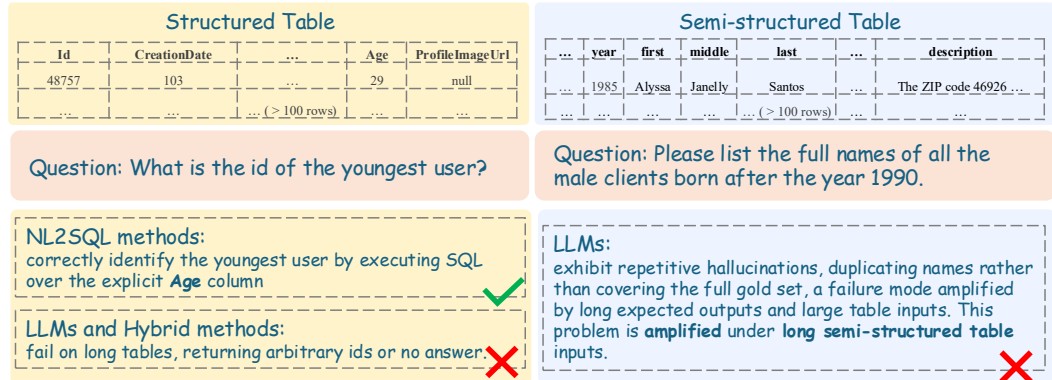

Figure 9: Case study: NL2SQL correctly finds the youngest user on a long structured table, while LLMs and hybrids produce spurious results.

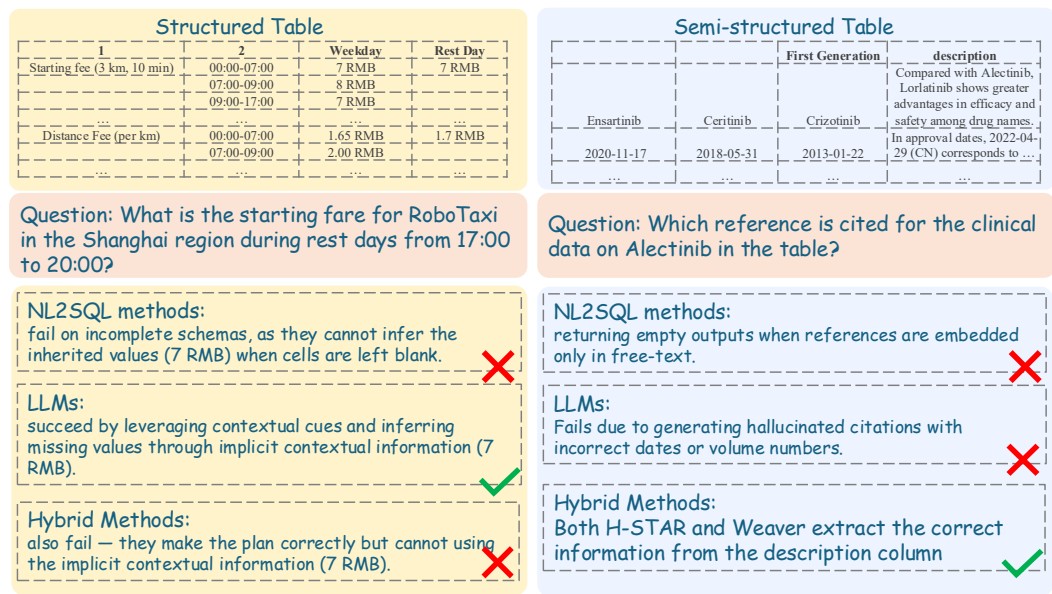

Figure 10: Incomplete schema example. LLMs can infer missing values while NL2SQL and hybrid methods fails (left). Hybrid methods can correctly extract the information while NLSQL and LLMs fail (right).

## H  USE OF LARGE LANGUAGE MODELS

We made limited use of large language models—specifically ChatGPT and GitHub Copilot—for language polishing and routine coding assistance (autocompletion, refactoring, and boilerplate generation). LLMs were not used to generate research ideas, experimental results, or claims. No non-public or sensitive data was shared with these tools. All suggested text and code were independently reviewed, tested, and edited by the authors, who remain responsible for the content.

Table 11: Structured vs. Semi-Structured accuracy across all splits. $\Delta$ (pp) = semi-structured minus structured.

| Split | Model | Structured Acc. (%) | Semi-Structured Acc. (%) | $\Delta$ (pp) |
|---|---|---|---|---|
| S1 | GPT-4o | 89.47 | 90.53 | +1.06 |
| | Gemini-2.5-Pro | 92.63 | 88.42 | -4.21 |
| | Gemini-2.5-flash | 90.53 | 88.42 | -2.11 |
| | Qwen3-235B | 85.26 | 89.47 | +4.21 |
| | LLM-NL2SQL | 87.37 | 62.11 | -25.26 |
| | XiYan | 78.95 | 26.32 | -52.63 |
| | H-STAR | 75.79 | 80.00 | +4.21 |
| | Weaver | 88.42 | 91.58 | +3.16 |
| S2 | GPT-4o | 71.60 | 63.31 | -8.29 |
| | Gemini-2.5-Pro | 81.07 | 73.37 | -7.70 |
| | Gemini-2.5-flash | 73.96 | 67.46 | -6.50 |
| | Qwen3-235B | 66.86 | 57.40 | -9.46 |
| | LLM-NL2SQL | 24.26 | 15.98 | -8.28 |
| | XiYan | 30.18 | 18.94 | -11.24 |
| | H-STAR | 88.17 | 81.66 | -6.51 |
| | Weaver | 86.39 | 76.33 | -10.06 |
| S3 | GPT-4o | 46.90 | 43.58 | -3.32 |
| | Gemini-2.5-Pro | 76.59 | 73.86 | -2.73 |
| | Gemini-2.5-flash | 61.24 | 63.07 | +1.83 |
| | Qwen3-235B | 37.62 | 38.07 | +0.45 |
| | LLM-NL2SQL | 74.77 | 48.18 | -26.59 |
| | XiYan | 73.79 | 34.94 | -38.85 |
| | H-STAR | 48.74 | 47.82 | -0.92 |
| | Weaver | 57.47 | 49.66 | -7.81 |
| S4 | GPT-4o | 53.85 | 48.72 | -5.13 |
| | Gemini-2.5-Pro | 60.76 | 59.49 | -1.27 |
| | Gemini-2.5-flash | 46.84 | 44.30 | -2.54 |
| | Qwen3-235B | 45.57 | 44.30 | -1.27 |
| | LLM-NL2SQL | 64.10 | 51.28 | -12.82 |
| | XiYan | 55.13 | 16.67 | -38.46 |
| | H-STAR | 28.21 | 23.08 | -5.13 |
| | Weaver | 53.85 | 55.13 | +1.28 |
| S5 | GPT-4o | 21.32 | 20.54 | -0.78 |
| | Gemini-2.5-Pro | 41.31 | 33.98 | -7.33 |
| | Gemini-2.5-flash | 5.41 | 5.81 | +0.40 |
| | Qwen3-235B | 18.92 | 17.05 | -1.87 |
| | LLM-NL2SQL | 62.55 | 37.98 | -24.57 |
| | XiYan | 62.79 | 27.52 | -35.27 |
| | H-STAR | 8.11 | 8.14 | +0.03 |
| | Weaver | 31.78 | 29.46 | -2.32 |
| M1 | GPT-4o | 53.40 | 50.82 | -2.58 |
| | Gemini-2.5-Pro | 82.44 | 78.45 | -3.99 |
| | Gemini-2.5-flash | 75.88 | 71.19 | -4.69 |
| | Qwen3-235B | 41.45 | 41.22 | -0.23 |
| | LLM-NL2SQL | 83.37 | 33.96 | -49.41 |
| | XiYan | 84.54 | 18.97 | -65.57 |
| M2 | GPT-4o | 19.88 | 16.87 | -3.01 |
| | Gemini-2.5-Pro | 35.54 | 29.52 | -6.02 |
| | Gemini-2.5-flash | 21.69 | 21.69 | 0.00 |
| | Qwen3-235B | 15.66 | 13.25 | -2.41 |
| | LLM-NL2SQL | 71.69 | 30.72 | -40.97 |
| | XiYan | 76.51 | 19.88 | -56.63 |

