# OpenReview forum: "Same Content, Different Representations: A Controlled Study for Table QA"
_ICLR.cc/2026/Conference — ICLR 2026 Poster_

### Official Review · Reviewer_DE2a · 2025-10-28

**Soundness:** 2
**Presentation:** 2
**Contribution:** 2
**Rating:** 4
**Confidence:** 2

**Summary:**

This paper presents the controlled study examining how table representation (structured vs. semi-structured formats) affects Table QA performance across different modeling paradigms. The paper introduces a pipeline that generates paired structured and semi-structured tables with identical content, enabling direct comparison of NL2SQL, LLM-based, and hybrid methods under controlled conditions. The study reveals consistent trade-offs.

**Strengths:**

1  The pipeline that generates semantically equivalent structured and semi-structured table pairs is reasonable.

2  The four-dimensional diagnostic framework (table size, joins, query complexity, schema quality) provides good insights into how different factors interact with representation formats.

3  The case studies and error analysis provide concrete examples

**Weaknesses:**

1 The verbalization process may not fully capture the complexity of naturally occurring semi-structured tables. Real-world tables often exhibit more irregular patterns and noise than the systematically generated semi-structured versions used in this study.

2 The paper provides limited theoretical analysis of why different representations affect model performance. More fundamental insights into the cognitive or architectural reasons for these differences would strengthen the contribution.

3 The paper could discuss the scale limitations in evaluation, for example, long tables with more than 100 rows.

**Questions:**

see weakness

---

> ### Author Response · Authors · 2025-11-22
>
> Thank you for your positive feedback about our benchmark's design and error analysis. We address each of your concerns:
>
> > **Weakness 1. The verbalization process may not fully capture the complexity of naturally occurring semi-structured tables. Real-world tables often exhibit more irregular patterns and noise than the systematically generated semi-structured versions used in this study.**
> - While we acknowledge that the verbalization strategy is simple, our goal in this paper is to conduct a *controlled* study of representation effects, in the spirit of recent controlled benchmarks such as HoloBench [1]. We deliberately start from a small set of simple, well-defined transformations to construct semi-structured versions, so that we can clearly attribute performance differences to the shift between structured and semi-structured formats. Notably, even under these minimal and systematically generated changes, the three method families already exhibit markedly different behavior, which is sufficient to reveal the core sensitivity we aim to study. We agree that real-world semi-structured tables can be more irregular and noisy. Introducing many such additional structural operations in the current benchmark would mix multiple factors at once and make it harder to isolate which transformation is responsible for each effect. We therefore treat richer, more realistic transformations as an important direction for follow-up work on top of our benchmark, rather than something we can comprehensively cover within the scope of a single paper.
>
>     [1] Holistic Reasoning with Long-Context LMs: A Benchmark for Database Operations on Massive Textual Data. ICLR 2025.
>
>
> > **2. The paper provides limited theoretical analysis of why different representations affect model performance. More fundamental insights into the cognitive or architectural reasons for these differences would strengthen the contribution.**
> - While analyzing results from the theoretical perspective is also interesting, our primary goal in this work is to build a controlled benchmark and systematically characterize empirical trends under representation shifts. So, the main paper focuses on accuracy rather than theory. We do provide some initial qualitative analysis in Sec. 5.6 and Appendix D, where we examine how representation shifts change failure modes for NL2SQL, hybrid, and LLM-based methods from the viewpoint of their characteristics. A fuller theoretical account of the underlying cognitive or architectural mechanisms would require additional tools such as controlled architectural ablations or probing studies, which we view as beyond the scope of this paper. We clarify in the revision that our contribution is to provide a solid empirical foundation and testbed for the kind of deeper analyses the reviewer suggests. We would also welcome any pointers to specific hypotheses or analytical tools that could help guide such analyses, and we would be happy to highlight these directions in our discussion of future work.
>
>
> > **3. The paper could discuss the scale limitations in evaluation, for example, long tables with more than 100 rows.**
> - As defined in Section 2.2, we treat tables with more than 100 rows as “long,” and Table 1 marks S4 and S5 precisely as the long-table splits, so all tables in S4/S5 fall into the >100-row regime mentioned by the reviewer. Our current benchmark focuses on tables that still fit into the context windows of current LLMs; extending the framework to ultra-long tables (hundreds or thousands of rows, likely requiring chunking or retrieval) is a natural direction for future work, and we will clarify this in the revision.
>
> We again appreciate your detailed review. We hope this response addresses your main concerns.

---

### Official Review · Reviewer_ZC3E · 2025-10-30

**Soundness:** 3
**Presentation:** 3
**Contribution:** 2
**Rating:** 6
**Confidence:** 4

**Summary:**

This paper presents the first controlled experimental study on a key yet underexplored problem in Table QA: the impact of table representation on model performance. It provides a comparative analysis of NL2SQL, LLM, and hybrid methods across different representation formats. The research is rigorously designed and supported by extensive experiments.

**Strengths:**

1. **Substantial Innovativeness**: The impact of table representation formats on model performance is an important yet systematically understudied topic.
2. **Comprehensive Evaluation**: The assessment covers multiple approaches—including NL2SQL, LLMs, and hybrid methods—across various datasets and evaluation dimensions.

**Weaknesses:**

1.  **Limited Table Representations Explored**: Although this study systematically investigates the impact of table representation formats on model performance, the variety of representations examined remains limited. Other transformations—such as row/column permutation, transposition, and modality conversion (e.g., to visual representations)—could be considered.
2.  **Room for Enhanced Table Diversity**: The current definition of short and long tables based solely on row count could be expanded. For instance, tables with very high column counts—common in fields like medical research—represent another important dimension of diversity.
3.  **Over-Simplified Table Transformation Methods**: Even when considering only structured and semi-structured tables, the transformation methods applied (e.g., textual serialization and cell merging) remain relatively basic. More advanced structural transformations could be explored.
4.  **Potential Data Insufficiency in Some Experiments**: The sample size in certain experiments (e.g., S4) may be insufficient for robust conclusions.
5.  **Recommendation: From Analysis to Decision Support**: Beyond data analysis, we recommend developing a decision tree to guide practitioners in selecting the most suitable method for specific scenarios.

**Questions:**

1.  **Additional Experiments on Schema Restoration**: Consider incorporating experiments that utilize methods like NameGuess[1] to restore damaged schemas, evaluating how such restoration impacts model performance.
2.  **Table Reconstruction Prior to Testing**: It may be beneficial to experiment with table reconstruction as a preprocessing step before applying the evaluation methods.
3.  **Mitigating LLM Judge Bias**: To address potential biases introduced by LLM-based evaluation, it is advisable to sample a subset of data for analysis—comparing LLM judgments with human scoring—to enhance the validity of the evaluation approach.

[1] Zhang J, Shen Z, Srinivasan B, et al. NameGuess: Column name expansion for tabular data[J]. arXiv preprint arXiv:2310.13196, 2023.

---

> ### Author Response · Authors · 2025-11-22
>
> Thank you for recognizing the innovativeness of table representation format analysis and our comprehensive evaluation. We address your main concerns:
> > **Weakness 1.Limited Table Representations Explored: Although this study systematically investigates the impact of table representation formats on model performance, the variety of representations examined remains limited. Other transformations—such as row/column permutation, transposition, and modality conversion (e.g., to visual representations)—could be considered.**
> - While the space of table representations is broad, our focus was on a practically informative subset of representation dimensions underexplored in prior work: structured vs. semi-structured layouts, table size, join complexity, and schema quality. These factors are highly consequential for Table QA pipelines, justifying our experimental design.
> - We omitted row/column permutations and related reorderings, as examined in earlier work [1] and our preliminary experiments, because they did not show systematic effects beyond those captured by our structured vs. semi-structured comparisons.
> - We acknowledge that visual or multimodal table representations are an exciting, distinct line of investigation, but they require different models (vision or multimodal LMs) and evaluation protocols. This is a natural extension of our framework, but not a clean fit for the current setup since we focus on the performance difference over structured and semi-structured tables, which has not been explored.
>
>     [1] The Mighty ToRR: A Benchmark for Table Reasoning and Robustness. arxiv 2025.
>
> > **2. Room for Enhanced Table Diversity: The current definition of short and long tables based solely on row count could be expanded. For instance, tables with very high column counts—common in fields like medical research—represent another important dimension of diversity.**
> -  Per the request, we additionally analyze **table width** on the BIRD-based splits (S1, S3, S4, S5). We bucket tables into three width levels: narrow (1–5 columns), medium (6–20), and wide (>20). The resulting test sets contain roughly the following number of instances:
>
>     **Table X: Width distribution on BIRD-based splits**
>
>     |Width bucket|# instances|
>     |--|--|
>     |Narrow (1–5) |455|
>     |Medium (6–20)|329|
>     |Wide (>20)|89|
>
>     For each method family we report accuracy separately for each width bucket and representation. The summary below shows the accuracies (in %) for structured vs. semi-structured tables:
>
>     **Table Y: Accuracy by table width (structured)**
>
>     | Model| Narrow | Medium | Wide |
>     |--|:-:|:-:|:-:|
>     | Qwen| 42.64  | 34.35  | 28.09 |
>     | Gemini-2.5| 51.87|43.47| 31.46 |
>     | GPT-4o| 52.75 | 37.80| 27.91 |
>     | NL2SQL|76.48|65.55| 57.30 |
>     | XiYan | 75.16| 64.94| 56.18 |
>     | Weaver  | 54.41| 44.07| 38.20 |
>     | H-STAR| 43.30| 32.01| 25.84 |
>
>     **Table Z: Accuracy by table width (semi-structured)**
>
>     | Model | Narrow | Medium | Wide |
>     |--|:-:|:-:|:-:|
>     | Qwen  |43.96| 31.91| 30.34 |
>     | Gemini-2.5| 54.29 | 41.34  | 31.46 |
>     | GPT-4o| 48.79  | 37.92  | 25.58 |
>     | NL2SQL  | 48.79  | 43.90  | 47.19 |
>     | XiYan | 24.18  | 35.06  | 43.82 |
>     | Weaver | 57.71  | 51.06  | 33.71 |
>     | H-STAR  | 44.84  | 32.52  | 21.35 |
>
>     Across all method families, accuracy generally decreases as tables become wider, with wide tables (>20 columns) being consistently more challenging than narrow ones. We will include these tables and a short discussion in the appendix during the rebuttal.
> We appreciate your critical suggestion and it makes our analysis more comprehensive.

---

> > ### Author Response · Authors · 2025-11-22
> >
> > > **3. Over-Simplified Table Transformation Methods: Even when considering only structured and semi-structured tables, the transformation methods applied (e.g., textual serialization and cell merging) remain relatively basic. More advanced structural transformations could be explored.**
> > - Our goal in this paper is to conduct a *controlled* study of representation effects, in the spirit of recent controlled benchmarks such as HoloBench [2]: we deliberately start from a small set of simple, well-defined transformations (textual serialization and basic cell merging) so that we can clearly attribute performance differences to the shift between structured and semi-structured formats. Even under these minimal changes, the three method families already exhibit markedly different behavior, which is sufficient to reveal the core sensitivity we aim to study. Introducing many additional, more complex structural operations (e.g., nested headers, grouped cells, extra contextual noise) would mix multiple factors at once and make it harder to isolate which transformation is responsible for each effect. We therefore treat richer transformations as an important direction for follow-up work on top of our benchmark, rather than something we can comprehensively cover in the scope of a single paper.
> >
> > [2] Holistic Reasoning with Long-Context LMs: A Benchmark for Database Operations on Massive Textual Data. ICLR 2025.
> >
> > > **4. Potential Data Insufficiency in Some Experiments: The sample size in certain experiments (e.g., S4) may be insufficient for robust conclusions.**
> > - The S4 split is designed as a **diagnostic** setting to cover all representation categories we study, rather than as a standalone large-scale benchmark. Each category in S4 still contains at least 78 queries. This scale is comparable to many controlled LLM benchmarks. For example, HoloBench [2] contains 90 questions in total, and in LiveBench [3] most individual tasks use just 50 queries. Our main empirical conclusions are supported by the larger splits; S4 is only used to probe specific categories more finely.
> >
> > [2] Holistic Reasoning with Long-Context LMs: A Benchmark for Database Operations on Massive Textual Data. ICLR 2025.
> >
> > [3] LiveBench: A Challenging, Contamination-Limited LLM Benchmark. ICLR 2025.
> >
> > > **5. Recommendation: From Analysis to Decision Support: Beyond data analysis, we recommend developing a decision tree to guide practitioners in selecting the most suitable method for specific scenarios.**
> > - We appreciate this practical suggestion and we agree that using visual guidance for decision support is a great idea since our analysis reveals clear patterns. For example,NL2SQL and hybrid pipelines tend to perform better on clean schemas and shorter, well structured tables, while LLM based methods are more robust on noisy or semi structured tables with incomplete schemas. In the revised version, we will turn these patterns into a concise decision tree for practitioners. The tree will branch on factors such as schema quality, table length and availability of supervision, and will lead to a concrete recommendation of NL2SQL, LLM or hybrid methods for each combination. This makes our findings directly usable as a method selection guide in real deployments.

---

> > > ### Author Response · Authors · 2025-11-22
> > >
> > > > **Question 1: Additional Experiments on Schema Restoration: Consider incorporating experiments that utilize methods like NameGuess[1] to restore damaged schemas, evaluating how such restoration impacts model performance. [1] Zhang J, Shen Z, Srinivasan B, et al. NameGuess: Column name expansion for tabular data[J]. arXiv preprint arXiv:2310.13196, 2023.**
> > > - We appreciate the reviewer for pointing out this relevant line of work. Our goal in this paper is to study how different QA methods behave under controlled representation shifts rather than to benchmark specific schema-restoration techniques. Methods such as NameGuess are highly complementary, but they introduce an additional preprocessing stage and effectively define a new end-to-end pipeline, which is beyond the scope of the current study. We note that our hybrid baselines (H-STAR and Weaver) already include schema-linking and header-interpretation mechanisms that implicitly mitigate some schema damage, and our results therefore partially reflect how such restoration-like components interact with different representations. We will mention schema-restoration approaches like NameGuess as a promising direction for future work building on top of our benchmark.
> > >
> > > > **2. Table Reconstruction Prior to Testing: It may be beneficial to experiment with table reconstruction as a preprocessing step before applying the evaluation methods.**
> > > - Since a preprocessing step usually helps in table reasoning tasks as you pointed out, the hybrid baselines we used incorporate non-trivial reconstruction-like preprocessing. For example, Weaver performs a pre-processing step that normalizes column formats, renames problematic headers, and augments the table with typed column descriptions before planning; H-STAR’s Table Extraction stage explicitly constructs a question-specific sub-table by selecting and reformatting relevant columns and rows. In our experiments these hybrid pipelines are noticeably more robust than pure NL2SQL baselines under damaged or incomplete schemas, so our results already partially capture the effect of such reconstruction-style components under different representations. Designing and benchmarking dedicated table-reconstruction modules would effectively introduce a new family of end-to-end pipelines, which we view as an orthogonal and valuable direction for future work building on top of our benchmark.
> > >
> > > > **3. Mitigating LLM Judge Bias: To address potential biases introduced by LLM-based evaluation, it is advisable to sample a subset of data for analysis—comparing LLM judgments with human scoring—to enhance the validity of the evaluation approach.**
> > > - Following the suggestion, we randomly sampled 100 evaluation cases, covering all datasets and baseline methods, and compared the LLM judge’s decisions with human annotations. The results show a 96% agreement rate. We will add this validation procedure in the appendix.
> > >
> > > We appreciate your detailed and constructive feedback. We believe that addressing your points has significantly improved the clarity and analytical depth of our paper.

---

### Official Review · Reviewer_kAHU · 2025-11-01

**Soundness:** 3
**Presentation:** 2
**Contribution:** 2
**Rating:** 4
**Confidence:** 3

**Summary:**

This paper presents a controlled study on how table representation, structured vs. semi-structured, affects model performance in Table Question Answering (Table QA). Using a verbalization pipeline that preserves table semantics, the authors compare NL2SQL, LLM-based, and hybrid approaches across varied settings of table complexity and query difficulty. Results show that structured inputs favor symbolic methods, semi-structured forms favor LLMs, and hybrid systems achieve the best balance, highlighting how representation choice fundamentally shapes reasoning accuracy and robustness.

**Strengths:**

- The study introduces a novel, well-controlled evaluation framework that isolates representation effects with strong methodological rigor. It provides comprehensive empirical evidence across datasets (BIRD, MMQA, TableEval), method families, and conditions, revealing consistent, interpretable trends.

- The verbalization pipeline and diagnostic splits are technically elegant and reproducible, offering a valuable benchmark for future Table QA research. The analyses are thorough, uncovering key trade-offs and failure cases that inform model and benchmark design.

**Weaknesses:**

The paper is largely empirical, offering extensive measurements but limited analytical explanation for why representational shifts yield performance divergence: What internal representations or attention dynamics contribute to this behavior? While the benchmark is comprehensive, it focuses mainly on static accuracy; are there analyses on error propagation, confidence calibration, or reasoning trace fidelity under different table formats?

The verbalization pipeline relies on LLM paraphrasing—how consistent is this process across seeds, and how might linguistic variation bias outcomes? The study primarily evaluates medium-scale models and standard datasets; can the results generalize to larger instruction-tuned LLMs or real-world multi-table enterprise data?

Finally, while comparisons are broad, statistical significance testing and variance reporting across multiple runs are limited. Could more rigorous uncertainty quantification strengthen empirical claims?

**Questions:**

See the Weaknesses above.

---

> ### Author Response · Authors · 2025-11-22
>
> We appreciate your careful and detailed review. We address each of your concerns:
> > **Weakness 1. The paper is largely empirical, offering extensive measurements but limited analytical explanation for why representational shifts yield performance divergence: What internal representations or attention dynamics contribute to this behavior? While the benchmark is comprehensive, it focuses mainly on static accuracy; are there analyses on error propagation, confidence calibration, or reasoning trace fidelity under different table formats?**
> - As you recognize that our benchmark is comprehensive, our main goal in this work is to build a controlled benchmark and to characterize performance trends under representation shifts across heterogeneous querying paradigms, which is why the paper focuses on static accuracy rather than theoretical analysis. Also, to understand when and why each baseline method struggles, we provide informative analysis in Sec. 5.6 and Appendix D where we examine how representation shifts change dominant failure modes and where in the pipeline errors occur. However, a unified treatment of internal attention dynamics, error propagation, calibration, and reasoning-trace fidelity is not technically feasible in our current setting for following reasons:
>     - Conducting a systematic error-propagation study would require comparable intermediate signals (e.g., schema-linking decisions, partial SQL plans, execution traces) and gold annotations for every stage in the pipeline. However, many baselines are black-box implementations that expose only final answers. Existing benchmarks also lack ground truth for intermediate steps. Thus, any such analysis would either cover only a small subset of methods or rely on ad-hoc proxies that are not comparable across paradigms.
>     - Most of the models in our study return only final outputs without calibrated probabilities or logits in a consistent format, which makes it impossible to compute meaningful confidence-calibration metrics uniformly across all baselines. Likewise, internal representation or attention-pattern analyses are not applicable to proprietary closed-source models.
>     - Reasoning traces vary greatly by architecture: NL2SQL and hybrid systems produce structured programs or multi-stage retrieval-execution pipelines, whereas LLMs generate free-form rationales that we do not always request. Designing a single notion of “trace fidelity” meaningful and fair to all these different trace formats would require a separate, task-specific set up.
> - We see our benchmark and results as early evidence pointing toward the need for finer-grained, multi-step annotations and for more transparent methodologies that make intermediate reasoning traces accessible.
>
> > **2.1 The verbalization pipeline relies on LLM paraphrasing—how consistent is this process across seeds, and how might linguistic variation bias outcomes?**
> - In our setup, LLMs are used only once to generate constrained verbalization templates. The prompts explicitly forbid introducing any additional information and supply the true column names and representative values, which keeps the semantics of the verbalized tables stable. We do not vary random seeds for this step; instead, we treat the resulting verbalized text as a fixed component of the benchmark. The final benchmark includes 1,628 instances drawn from several existing table QA datasets, spanning diverse question types and table schemas. Given this scale and heterogeneity, we do not expect minor stylistic choices in the templates to introduce systematic biases favoring any particular model family.
>
> > **2.2 The study primarily evaluates medium-scale models and standard datasets; can the results generalize to larger instruction-tuned LLMs or real-world multi-table enterprise data?**
> - Regarding generalization, our evaluation already includes **large-scale instruction-tuned LLMs** (GPT-4o), and in the revision we will include the result of Gemini-2.5-Pro. The results are summarized below:
>     | Dataset | Structured Acc | Semi-Structured Acc |
>     |-|-|-|
>     | S1| 92.63% | 88.42%|
>     | S2| 81.07%| 73.37% |
>     | S3| 76.59%| 73.86%|
>     | S4| 60.76%| 59.49%|
>     | S5| 41.31%| 33.98%|
>     | M1| 82.44%| 78.45%|
>     | M2| 35.54%| 29.52%|
>
>     Gemini-2.5-Pro shows the same representation-sensitive trend as our other models: its accuracy consistently drops when we move from structured to semi-structured tables. So our conclusions are not restricted to medium-scale models.
> -  As for the real-world multi-table data, our benchmark already covers many properties of real-world multi-table enterprise settings, since the diagnostic splits span BIRD, MMQA, and TableEval with diverse table sizes, join patterns, and imperfect schemas.

---

> > ### Author Response · Authors · 2025-11-22
> >
> > > **3. while comparisons are broad, statistical significance testing and variance reporting across multiple runs are limited. Could more rigorous uncertainty quantification strengthen empirical claims?**
> > - While uncertainty quantification is important, running multiple independent rounds is prohibitively expensive, especially for API-based LLMs, given the scope of our evaluation. Instead, we rely on large sample sizes (hundreds to over a thousand test instances per split). The representation-sensitive effects we report are large and consistently directed across datasets and models. Given this, the sampling error of accuracies is small relative to the effect sizes, making the current empirical evidence sufficient for our claims.

---

### Official Review · Reviewer_9Uki · 2025-11-01

**Soundness:** 3
**Presentation:** 4
**Contribution:** 3
**Rating:** 6
**Confidence:** 3

**Summary:**

The authors perform a controlled study to examine how table representation affects model performance. The diagnostic benchmark they created has structured tables, but also has semi-structured variants, which are produced by selecting columns to change into free-text, reconstructing templates and serialize the updated table. They look into five key factors: structured vs semi-structured inputs, short vs long tables, multi-table alignment vs join operation tasks, simple lookups vs multi-step compositional queries and noisy schemas. The method families they test on are LLM-based, NL2SQL methods and hybrid methods. They generally find NL2SQL models to be better with structured tables and hybrid models to be more balanced with robustness across formats and peak accuracy.

**Strengths:**

S1. The paper explores various facets of the table representation, which are reasonable and comprehensive directions.
S2. The paper is generally well structured and organized.

**Weaknesses:**

W1. Some of the claims don't seem to match the actual results. For instance, on lines 284-286, the claim is that "LLMs degrade modestly" and "hybrids lead" for RQ1. However, based on Table 2, the drop % for LLM methods is similar or even much lower than the Hybrid ones. It doesn't seem that hybrid models lead in that case.

**Questions:**

See W1.

---

> ### Author Response · Authors · 2025-11-22
>
> Thank you for your feedback. We would like to provide the clarifications about the concern you raised.
>
> > **Weakness 1. Some of the claims don't seem to match the actual results. For instance, on lines 284-286, the claim is that "LLMs degrade modestly" and "hybrids lead" for RQ1. However, based on Table 2, the drop % for LLM methods is similar or even much lower than the Hybrid ones. It doesn't seem that hybrid models lead in that case.**
>
> - We agree that in Table 2, the drop percentages for LLM and hybrid methods are close and in some cases, LLMs show a smaller drop. Our statement about hybrid pipelines “leading” was meant to capture a broader pattern observed across multiple experiments and conditions. In particular, hybrid methods tend to be more robust as tables get longer, queries more compositional, or schemas noisier. In contrast, LLM-only methods perform well particularly when supervision is limited and the queries are relatively short and simple.
>
> - To avoid confusion, we will revise the text to more accurately reflect these nuances and avoid overgeneralizing from a single metric. Additionally, we plan to summarize the findings as a decision tree that provides recommendations for NL2SQL, LLM, or hybrid approaches under different settings.

---

### Meta-Review · Area_Chair_fGfh · 2026-01-06

**Summary:**

This paper introduces a diagnostic benchmark to evaluate the effect of table representation in table QA over structured and semi-structured tables. The paper uses an LLM-based verbalization pipeline which creates semi-structured tables from structured ones, allowing a direct comparison on paired tables. It evaluates LLM-based, NL2SQL, and hybrid methods, and finds some consistent trends (e.g. NL2SQL methods are sensitive to table structure).

Reviewers generally found the evaluation comprehensive, exploring multiple possible table representations across multiple datasets. One reviewer found the verbalization pipeline "technically elegant" and multiple reviewers found it to be well-controlled and reproducible. The paper and its diagnostic benchmark should be a useful contribution for research in table QA.

**Reviewer Concerns:**

One main reviewer concern was that the verbalization table transformations are simple and so may not capture real-world semi-structured tables. I felt like the author response -- that simplicity allows for a controlled study, and (at least with this current set of models) is sufficient to produce variations in performance across the three families of methods -- was convincing.

Another main concern was that there is limited analysis of *why* the methods have differing performance across the different formats. I do agree that this is a limitation of the paper, but the paper does provide guidance on which method family to use based on properties of the table and this should still be useful to practitioners.

A few reviewers also raised concerns about statistical significance. Given the large differences in accuracy between some of the methods, and the large sizes of the datasets, I'm less concerned about this -- but the authors should still report statistical significance (and I'm a bit surprised that they didn't in the author response).

**Reviewer Scores:**

I expect that kAHU might have raised their score to a 6 given that the responses about verbalization robustness. The other three reviewers probably would have kept their scores the same, given that they were either already positive (9Uki, ZC3E) or had more concerns (DE2a) that weren't fully addressed by the response. So the overall scores might have been 6 / 6 / 6 / 4.

---

### Decision · Program_Chairs · 2026-01-26

Accept (Poster)